# The Neural Response Process of Cognitive Decision Making: An ERP Study

**DOI:** 10.3390/brainsci13040648

**Published:** 2023-04-11

**Authors:** Xiaolei Gao, Lei Gao, Yutong Li, Xue Sui

**Affiliations:** 1School of Education, Tibet University, Lhasa 850000, China; 2School of Psychology, Liaoning Normal University, Dalian 116029, China; dearliyutong@163.com

**Keywords:** cure rate, risk, cognitive decision making, ERP

## Abstract

Cognitive decision has the basic characteristics of risk avoidance and benefit seeking. To explore the neural response process of cognitive decision making, we asked 32 undergraduates to make a decision on whether to accept a specific treatment option with a certain cure rate and a certain risk rate while recording their electrical brain responses. The results showed that more participants chose the treatment option with a high cure rate and moderate or low risk. Compared with low and high risk, medium risk produced greater N1 and smaller P300. Low risk produced larger LPP than the moderate risk in the left hemisphere. The right prefrontal region appeared to have a smaller LPP for low risk than for high risk. The results suggest that individuals prioritize risk when making cognitive decisions. In addition, in medium-risk conditions, solution integration is more difficult. The effect of benefit size appears at the late stage of cognitive decision making and adjusts the effect of risk. These results support the satisfaction principle of decision making.

## 1. Introduction

Individuals face decisions throughout their lives. Some decisions are simple and straightforward, which can be decided directly based on maximizing the benefits they receive. However, most of the decisions, including cognitive decisions, are dilemmas where costs are weighed against benefits. Cognition decisions often rely on a comparison of uncertain costs and potential benefits [1]. We require a decision-making framework to judge and quantify both costs and outcomes of relevance and criteria to determine the optimal decision [2]. Outcomes and costs are potentially considered sequentially rather than simultaneously. There may be a trade off between the two. Currently, many types of research primarily focus on economic decision making and have yielded considerable fruitful achievements [3,4,5]. Researchers used investment decision tasks to investigate how consistent the two classes of models (utility-based models and risk–return models) are with the neurobiological processes underlying investment decisions. They provide evidence that risk–return models describe the neural processes underlying investment decisions well [3]. This also illustrates the dependence of cognitive decision making on risk assessment.

The COVID-19 outbreak in recent years has raised significant health concerns [6,7]. However, there is relatively little research on health and treatment decisions [8,9]. Similar to economic decisions, each treatment option in a decision has the probability of cure and the risk of death [10,11]. Generally, people magnify the importance of costs in the decision making and make conservative decisions instead [12]. In analogy, treatment compliance decreases as people abandon treatment with high risk. However, there is a great deal of specificity in the choice of risky treatment options when the body has a disease because there’s a fundamental risk involved here, which is that the disease gets worse.

Although there are no explanatory models for treatment decisions, some existing theoretical models can be used as foundations and references. Expected Utility theory is based on the hypothesis of rational economic humans, which regards humans as rational individuals. In order to maximize benefits (i.e., maximum utility), rational individuals weigh the conflicting options between gains and losses and choose the most optimal option [13]. Expected value maximization assumes that payoffs are multiplied against corresponding probabilities as a utility value [14]. The principle of decision satisfaction holds that in actual decision making, human rationality is limited; individuals follow the principle of satisfaction rather than the principle of optimality [15]. Value decision theory, on the other hand, explains the decision-making process from the perspective of objective value. According to value decision theory, people first assign value to different options and then make a comparison to choose the option that they think has the highest value [16].

In order to investigate the pattern of cognitive decisions in health and treatment situations, some studies simplified complex situations and conducted corresponding studies in laboratories. During the experiment, participants were asked to imagine they had a disease. The symptoms of this disease and the negative effects of non-treatment were also described. Additionally, participants were informed that there was a drug that effectively cures but also has side effects. In the formal experiment, cure rate and risk probability were simultaneously presented to the participants, and then they determined whether to accept the treatment [17]. This study revealed that patients’ adherence to treatment increased as their health literacy increased. Thus, it is recommended that the health literacy levels of the patients be raised through effective interventions to ensure better adherence to treatment. Of course, no treatment option is a 100% cure, nor is it 100% death free. Some options even have direct adverse effects. Some studies found that changes in decisions largely rely on the magnitude of side effects [18]. For instance, Felder reported that as risk probability increased, people gradually chose to abandon treatment [19], suggesting that treatment adherence decreased as side effects increased. Similarly, in a study by Bruce et al., they also found that people’s treatment compliance decreased with the increase of risk probability. When compared to adherent patients, nonadherent patients significantly devalued treatment efficacy and inflated treatment risk [10]. In addition, they also revealed that the cure rate played an important role in treatment decisions, suggesting that decision processes considered not only risk probability but also cure rate. People are more likely to receive treatments with lower risk probability and higher cure rates. In another study conducted by Bruce et al., they manipulated mild levels of risk probability and further explored the influence of cure rate, risk probability, and the severity degree of the side effects on treatment decisions [20]. The results were that the cure rate decreased, and the probability of side effects increased; patients were less likely to take the drug.

While such studies clearly suggest that cure rate and risk probability as the most important factors affecting treatment decisions; however, little research has been carried out to explore the neural responses of the influence of cure rate and risk probability on treatment decision making. Therefore, this study uses the existing EEG components used in decision-making research to explore this problem.

It is generally believed that the decision process is closely correlated with the N1, P300, and LPP components [21,22,23]. The N1 component is a negative wave peaking approximately 100–150 ms after stimulus onset [24]. It has been reported that the N1 is sensitive to selective attention processing at the early stage of decision making. Some studies have revealed that greater N1 amplitudes are elicited when individuals pay more attention to the valence of the choice item [21]. On the other hand, the P300 is a typical ERP component in decision making. It is a positive wave peaking roughly 300–500 ms. Increased P300 amplitudes are thought to reflect stronger motivational/emotional salience of outcomes [25]. The research found the P300 responses to division schemes were affected not only by the type of unequal offers but also by whom the property was initially assigned to [22]. The P300 is sensitive to top-down controlled processes. Moreover, the P300 waveform appears in response to active engagement in the detection of task-relevant target stimuli. The P300 waveform objectively measures large-scale neuronal network functioning and working memory processes [26]. In addition, the LPP is a positive deflection often observed beginning around 300–400 ms after the onset of the stimuli with a duration of several hundred milliseconds. Some studies have demonstrated that motivationally significant stimuli, such as emotional stimuli, in contrast to neutral stimuli, lead to larger LPP amplitudes [20,21]. In moral judgment, the LPP has been found to reflect the continuous attention to social or emotional salient information [27].

In this study, participants were asked to choose the option with a certain risk and a certain cure rate, and the EEG response was recorded. The neural mechanism of decision making has been studied. The brain waves related to decision making are N1, P300, and LPP. We predicted that: (1) subjects would give priority to the probability of risk, and the probability of risk increases and the likelihood of treatment options being chosen decreases, which is consistent with Bruce et al.’s studies [10,20]; (2) The effect of risk probability was regulated by the cure rate. Under the condition of specific risk probability, the cure rate increasing results in treatment options more likely being chosen; (3) The choice of treatment is closely related to the disease status. High-risk acceptance is low; low-risk acceptance is high. However, moderate risk, influenced by the possibility of disease progression, will receive more attention than low risk, with a larger N1 amplitude. (4) In late integration, compared with high risk, low-risk decision integration is easier, and the P300 and LPP amplitude is smaller. While the medium risk obtains more attention early, the LPP amplitude induced by the medium risk is smaller than that of the low risk.

## 2. Materials and Methods

### 2.1. Participants

To ensure sufficient statistical power, the required sample size was calculated by a power analysis based on the predicted effect using G* Power 3.1.9.2. The experiment was a within-factors design with repeated measures. We predicted a medium size (*f* = 0.25) with 82.26% actual power at the 0.05 significance level, the required sample size was at least 15 individuals for the experiment. We put up a small notice on the campus public notice board to recruit participants. College students signed up randomly by scanning QR codes. Thirty-two healthy college students (20 females) from 19 to 30 years (*M_age_* = 22.50; *SD* = 3.75) participated in this study. All participants were right-handed, possessed normal or corrected-to-normal vision and reported no neurological or psychiatric history. Each participant signed a written consent before taking part in a hypothetical treatment decision-making task. This experiment was approved by the Institutional Review Board at Tibet University. The methods were carried out in accordance with the Declaration of Helsinki [28]. The lead author Xiao-lei Gao and the corresponding author Xue Sui are both members of the Eye Movement Psychology Research Committee of the Chinese Psychological Society. They both attached great importance to using eye movement technology to solve the problem of language cognitive processing mechanisms. Therefore, a good cooperative relationship was established several years ago. In recent years, apart from continuing to pay attention to the cognitive processing of language, they have also begun to pay attention to other problems of cognitive processing in the brain. In this process, their collaborative research relationship has become closer than ever.

### 2.2. Materials

Cure rate and risk probability were both divided into three levels: low (10–39%), medium (40%–69%), and high (70%–99%). In terms of Abidi’s experiment [29], we had nine combinations of cure rate and risk probability, including Lc-Lr (low cure rate and low risk), Lc-Mr (low cure rate and medium risk), Lc-Hr (low cure rate and high risk), Mc-Lr (medium cure rate and low risk), Mc-Mr (medium cure rate and medium risk), Mc-Hr (medium cure rate and high risk), Hc-Lr (high cure rate and low risk), Hc-Mr (high cure rate and medium risk), and Hc-Hr (high cure rate and high risk). In our experiment, ten values are randomly selected from each level interval, and combined without repetition, resulting in 90 combinations.

In the preliminary experiment, it was found that there were too many combinations and too long of a response time (more than 3000 ms) for each trial and too many EEG data artifacts. Therefore, we simplified the experimental material. Three fixed values were chosen to represent three levels (low: 36%, medium: 64%, and high: 98%). We combined various levels of risk probability and cure rate and obtained nine combinations. Each combination was presented in a top-down arrangement, where the risk and cure rate were presented at the upper side for one time, respectively. Therefore, there were 18 images as stimuli.

In other words, the materials were images showing the probability of risk and the probability of cure. Each image was a black background, and the combination was written in white color with song font in the center. The whole experiment consisted of 360 trials, including 20 repetitions of each image. Figure 1 showed a stimulus image with “36% 治愈率 and 98% 风险” (in English: 36% cure rate and 98% risk) written in the center.

### 2.3. Apparatus and Procedure

The participants were tested individually. Each participant sat in a comfortable chair in front of a computer monitor (resolution, 1920 × 1080 pixels; refresh frequency, 65 Hz). The experimental stimuli were presented in the center of the screen at a distance of 65 cm from the participant’s eyes. The visual angle of a word was approximately 0.88°.

Before the formal experiment, participants were informed to suppose that they were suffering from a serious disease. A treatment option would be presented on the screen. Each treatment included specified values of cure rate and risk probability. The cure indicates improvement or even complete recovery of health as a result of treatment. In contrast, the risk is the probability that treatment will result in deterioration or even death. Subjects were informed to evaluate the treatment option based on cure rate and risk probability and then to express whether they would accept this treatment or not by pressing buttons. If they choose it, press J; if they give up, press F (balance between subjects).

The experiment started with 5 practice trials and was followed by 4 blocks, each block including 90 trials. Appropriate rests were arranged between blocks. In each trial, a fixation appeared in the center of the screen for 500–900 ms; subsequently, the stimuli were presented for 3000 ms, then a screen followed with an instruction for participants to respond to (F or J in keyboard pressed for acceptance/rejection responses). After that, a blank screen was presented for 1000 ms at the end of the trial. A single trial is illustrated in Figure 2. The entire experiment took approximately 30 min to complete for each participant.

### 2.4. EEG Recording and Analysis

The electroencephalogram (EEG) signals were recorded using a 64-channel Brain Products system. The electrodes were placed according to the extended 10–20 system. The EEG record was referenced online against the FCz site and was grounded at the FPz site. The vertical electrooculogram (EOG) was recorded to identify blink artifacts. All electrode impedances were kept below 5 kΩ for the duration of the experiment.

Data were sampled at 500 Hz and filtered by 0.1 Hz–30 Hz (slope 24 dB/oct). The EEG data were analyzed by BrainVision Analyzer 2.0 (Brain Products GmbH). Data were re-referenced offline to mastoids (TP9 and TP10 electrodes). Movement and drift artifacts were manually rejected by individual raw data inspection. Blink artifacts were corrected by independent component analysis. Data were segmented in epochs from 200 ms pre-stimulus onset to 800 ms post-stimulus onset and were baseline corrected using data obtained from −200 ms to 0 ms.

Stimulus-locked ERPs were averaged from usable trials. All conditions retained at least 92% of usable trials. Single-subject averages were computed for each experimental condition. ERPs were quantified by measuring mean amplitudes in three latency intervals: the N1 (100~200 ms), the P300 (200~350 ms), and the LPP (360~600 ms). To test for ERP effects, estimates of the ERP were obtained in nine topographical clusters by averaging across corresponding electrodes (Figure 3). There were three midline clusters: anterior area (Fpz, Fz, and FCz), central area (Cz and CPz) and posterior area (Pz, POz, and Oz). The other six clusters were anterior-right area (Fp2, AF4, AF8, F2, F4, F6, F8, FC2, FC4, FC6, FT8, and FT10), anterior-left area (Fp1, AF3, AF7, F1, F3, F5, F7, FC1, FC3, FC5, FT7, and FT9), central-right area (C2,C4, C6, T8, CP2, CP4, CP6, TP8, and TP10), central-left area (C1,C3, C5, T7, CP1, CP3, CP5, TP7, and TP9), posterior-right area (P2, P4, P6, P8, PO4, PO8, and O2), and posterior-left area (P1, P3, P5, P7, PO3, PO7, and O1).

For each component, we performed 3 × 3 × 3 × 3 repeated measures ANOVAs with risk probability (low, medium, and high), cure rate (low, medium, and high), anteriority (anterior, central, posterior), and laterality (left, middle, and right). The sphericity assumption was evaluated using Mauchly’s test, and the Greenhouse–Geisser correction for the degrees of freedom was used in cases of non-sphericity. The Bonferroni correction was used to correct for multiple post hoc comparisons. We only reported significant main effects and the interaction of risk probability and cure rate, as well as significant interactions of topographic factors with at least one experimental factor. The effect size for the statistically significant factors was estimated using partial eta squared (η_p_^2^).

## 3. Results

Five subjects were excluded because of excessive EEG artifacts, that at least 15 usable trials per condition were satisfied for analysis [30]. In sum, 27 subjects were included in the analyses.

### 3.1. Behavioral

The descriptive result was presented in Figure 4. The proportion of acceptance responses was analyzed by a 3 (risk probability: low, medium, and high) × 3 (cure rate: low, medium, and high) repeated measures ANOVA, with risk probability and cure rate as two within-participant factors. The results found a significant main effect of risk probability, *F* (2, 52) = 53.02, *p* < 0.001, η_p_^2^ = 0.67. The low-risk probability produced the largest acceptance proportion, followed by the medium-risk probability, and the high-risk probability generated the smallest acceptance responses, *ps* < 0.001. The main effect of the cure rate reached statistical significance, *F* (2, 52) = 71.37, *p* < 0.001, η_p_^2^ = 0.73. People were more likely to accept treatments with the high-cure rate compared with those with a medium-cure rate, and the lowest acceptance proportion was elicited by the low cure rate, *ps* < 0.001.

We also observed a significant interaction of risk probability and cure rate, *F* (2.19, 56.95) = 3.93, *p =* 0.022, η_p_^2^ = 0.13. In the low-cure rate condition, larger acceptance proportion was produced by the low-risk probability, respectively, compared with the high-risk probability, 0.335, 95% CI [0.120, 0.550], *p* = 0.001, and the medium risk probability, 0.254, 95% CI [0.06, 0.447], *p* = 0.007. There was no significant difference between the medium- and the high-risk probability, *p* = 0.25. In the high-cure rate condition, the high-risk probability produced a smaller acceptance proportion compared with the low-risk probability, −0.334, 95% CI [−0.548, −0.120], *p* = 0.001, and the medium risk probability, −0.269, 95% CI [−0.464, −0.075], *p* = 0.004. The difference between the low and the medium risk probability was not found, *p* = 0.25. In the medium-cure rate condition, the low-risk probability generated a larger acceptance proportion compared with the high-risk probability, 0.600, 95% CI [0.415, 0.785], *p* < 0.001, and the medium-risk probability, 0.375, 95% CI [0.172, 0.578], *p* < 0.001. Moreover, the medium-risk probability produced a larger acceptance probability than high-risk probability, 0.225, 95% CI [0.064, 0.386], *p* = 0.004.

### 3.2. Neurophysiological Results

An overview of the neurophysiological data was shown in Table 1, Table 2 and Table 3. Amplitude values were averaged in different time windows. The grand average ERPs of selected electrodes in different clusters are presented in Figure 5, Figure 6 and Figure 7.

#### 3.2.1. N1

The results of the rmANOVA showed a significant main effect of risk probability, *F* (2, 52) = 3.864, *p* = 0.027, η_p_^2^ = 0.129. The medium-risk probability elicited larger N1 amplitudes than the low-risk probability, *p* = 0.03, and then the high-risk probability, *p* = 0.02. There was no difference between the low and the high-risk probability, *p* = 0.70. Moreover, there were no significant main effects or interactions with risk probability and cure rate or interactions of topographic factors with at least one experimental factor in these nine clusters, all *ps* > 0.159.

#### 3.2.2. P300

From the results of the rmANOVA, it revealed a main effect of risk probability, *F* (2, 52) = 3.734, *p* = 0.031, η_p_^2^ = 0.126. The low risk probability elicited larger P300 amplitudes than the medium risk probability, *p* = 0.036. The high risk probability elicited larger P300 amplitudes than the medium risk probability, *p* = 0.023. The difference between the low and the high risk probability was not significant, *p* = 0.023. In addition, there were no significant main effects or interactions were found, all *ps* > 0.083.

#### 3.2.3. LPP

We observed a significant interaction between risk probability and anteriority, *F* (2.936, 76.339) = 3.291, *p* = 0.026, η_p_^2^ = 0.112. Specifically, the low-risk probability elicited larger LPC amplitudes than the medium-risk probability in the left electrode cluster, 0.562, 95% CI [0.066, 1.057], *p* = 0.028. Other significant difference about risk probability conditions in any clusters was not observed, *ps* > 0.07.

Moreover, a significant three-way interaction between risk probability, anteriority and laterality was found, *F* (4.306, 111.966) = 4.752, *p* = 0.001, η_p_^2^ = 0.155. The low-risk probability elicited smaller LPP amplitudes than the high-risk probability in the anterior-right electrode cluster, −0.390, 95% CI [−0.724, −0.055], *p* = 0.024. The low-risk probability elicited larger LPP amplitudes than the high-risk probability in the posterior-right electrode cluster, 0.431, 95% CI [0.034, 0.829], *p* = 0.035, and the posterior-left electrode cluster, 0.439, 95% CI [0.030, 0.849], *p* = 0.036. The low-risk probability elicited larger LPP amplitudes than the medium-risk probability, which was found in the posterior-right electrode cluster, 0.578, 95% CI [0.134, 1.022], *p* = 0.013, the posterior-middle electrode cluster, 0.528, 95% CI [0.008, 1.048], *p* = 0.047, the posterior-left electrode cluster, 0.573, 95% CI [0.082, 1.065], *p* = 0.024, and central-left electrode cluster, 0.541, 95% CI [0.053, 1.029], *p* = 0.031.

Another significant three-way interaction between risk probability, cure rate, and anteriority were also observed, *F* (4.124, 107.234) = 2.589, *p* = 0.039, η_p_^2^ = 0.091. In the low-cure rate condition, the low-risk probability elicited larger LPP amplitudes than the medium-risk probability in the anterior electrode cluster, 0.725, 95% CI [0.084, 1.365], *p* = 0.028, and central electrode cluster, 0.764, 95% CI [0.036, 1.492], *p* = 0.04. In the medium cure rate condition, the high-risk probability evoked larger LPP amplitudes than the low-risk probability in the anterior electrode cluster, 1.018, 95% CI [0.014, 2.022], *p* = 0.047. In contrast, the low-risk probability produced larger LPP amplitudes than the medium-risk probability in the posterior electrode cluster, 0.719, 95% CI [0.197, 1.241], *p* = 0.009.

## 4. Discussion

In this study, by controlling the benefits and risks of the treatment option, the dependence degree of cognitive decision-making on the risks and benefits of the treatment option and its neural response was investigated. Behavioral results showed that acceptance proportions decreased with increasing risk probability and increased with increasing cure rate. We also found that risks and benefits have an interactive effect on the acceptance proportion. Specifically, in the low-cure rate, acceptance proportion for the low risk is larger than that for the medium and high risk. In the medium-cure rate, the acceptance proportion for the low risk is significantly larger than that for the medium risk probability, and acceptance proportion for the medium risk probability is obviously higher than that for high-risk probability. In the high-cure rate, the acceptance proportion for the low- and medium-risk probability is higher than that for high-risk probability. In other words, as the cure rate increases, the acceptance proportion of moderate risk increases gradually.

The behavioral results indicate that subjects are more willing to accept a low-risk treatment option. As the risk increases, the requirement for a cure rate is higher. This is consistent with previous findings, which found that subjects tended to reject the drug with high side effects and low efficacy [29]. The awareness of high risk is closely related to the health literacy of the subjects in this study. Health literacy plays an important role in treatment adherence in patients with chronic diseases [17]. Drugs are supposed to be effective, with no risk or very little risk. Therefore, people pay more attention to the risk. The research found that treatment propensity increases with the probability of death but can decrease with the severity of illness [19]. It can be seen that risk is a very important factor affecting decision making. Some researchers believe that treatment decision involves participants weighing the risks and benefits of medication [10,20]. In an ideal world, people would like to choose an option with as few risks as possible and as many benefits as possible. Nevertheless, for the most part, it is not that ideal, and you have to balance the risks against the benefits. Participants discount the subjective value of the reward as the likelihood of its receipt decreases. In other words, the increase in risk, the subjective value of the benefits from treatment decreased. Therefore, individuals give priority to risk when they make treatment decisions. When they can accept the harm caused by risks, they will consider the cure rate and then choose whether to accept this treatment.

We propose the threshold point hypothesis (TPH) to explain the behavior results. First, in individual experience, there is a threshold point for both risk and benefit, which varies from person to person. Second, the two dimensions of risk and benefit are at the threshold point, then the probability of choice and abandonment is close, without choice bias. Third, if one dimension is on the threshold point and the other deviates from the threshold point, the decision directly depends on the deviating direction. Fourth, when both dimensions deviate from the threshold point, options with low risk and high-cure rates are more likely to be accepted. Options with high risk and low-cure rates are more likely to be rejected. Options with high risk and cure rates are more likely to be rejected; options with lower risk and cure rates are more likely to be accepted. The neural mechanism can be analyzed from the EEG results.

EEG results showed that the medium risk condition elicited larger N1 amplitudes compared with the low and the high-risk condition. The N1 component is related to early selection attention [31]; individuals’ attention to options increases, and N1 amplitude increases [21]. There are differences in N1 amplitudes among the three risk conditions, maybe indicating that risks with different probabilities have different meanings for individuals. Low risks are easier to accept, and high risks are easier to reject. Moderate risk is qualitatively different. There are many factors to weigh in the medium risk. At this point, decisions need to be made in conjunction with other factors, such as the cure rate. Therefore, the increase in N1 amplitude may reflect increased early attention to particular risk attributes.

The present study also found that the P300 amplitude produced by medium risks was smaller than that of low and high risks. The amplitude accounts for the amount of attentional resources allocated to the task [32]. The P300 component is related to decision evaluation [33,34,35]. This component reflects motivational significance induced by the stimulus. When the motivation intensity of the individual to the stimulus increases, the amplitude of P300 increases. Moreover, diseases of old age also led to a decrease in the P300 amplitude, indicating insufficient cognitive activity [26]. In this study, different risk conditions are presented separately. Subjects were more likely to accept the low-risk option, and they were more likely to reject the high-risk option. Moderate risk requires patience, calm judgment, and decision making. In addition, the increase of the P300 amplitude is also related to the emotional valence of stimulus materials [36]. Compared with the medium risks, the low and high risks caused the subjects to produce a stronger emotional response, and the P300 amplitude of the low and high risks was larger than that of the medium risks.

In this study, low risks produced larger LPP amplitude than medium risks and high risks in posterior brain regions. This suggests that the selection of low-risk options is supported by late positive components. LPP waves are widely distributed in the anterior and posterior areas of the scalp [23]. In this study, LPP differences appeared in the posterior brain region, which was consistent with previous findings. LPP amplitude changes are associated with motivationally significant stimuli [37]. Compared with neutral stimuli, positive, and negative emotional stimuli produce larger amplitude LPP [23,25]. In the process of purchasing goods, the stronger an individual’s positive emotions are, the greater the fluctuation of LPP will be [38]. In this study, compared with medium and high risks, low risks mean less harm to the individual body, and individuals are more likely to produce positive emotions and induce greater LPP amplitude. Therefore, a larger amplitude LPP with low risks was observed in the posterior brain regions than with medium and high risks.

We also found that low risks produced smaller amplitudes of LPP than high risks in the right prefrontal region, while low risks produced larger amplitudes of LPP than medium risks in the left prefrontal region. The human frontal cortex is asymmetrically involved in motivation and emotional processing. The left frontal cortex is associated with positive emotions and approach motivation, while the right frontal cortex is associated with negative emotions and avoidance motivation [39]. In a lexical experiment describing “good” and “bad” concepts, it was found that positive valence words produced a larger amplitude left-frontal LPP, while negative valence words produced a larger amplitude right-frontal LPP [40]. This phenomenon was also observed in this study, providing experimental evidence that the human frontal cortex is asymmetrically involved in emotional processing. In addition, this study also found that for low risks, both low and high benefits produced greater amplitude LPP than those with medium benefits. The effect of risks on treatment options is moderated by the cure rate. This was consistent with behavioral results. When making treatment decisions, participants mainly considered risks. Individuals should consider making decisions based on the cure rate only if the harm from risks is acceptable.

The behavioral and EEG results of this study support the satisfaction principle of decision making. First of all, treatment decision-making does not adopt the optimal principle, and people are bounded rational. Individuals do not make decisions based solely on the cure rates but weigh risks and benefits to choose a satisfactory treatment. Secondly, individual decisions are based on subjective goal values. Treatment with different combinations of benefits and risks have different target values for individuals. By comparison, individuals choose the option that has the greatest value for them. The acceptance degree of the option was significantly polarized among the subjects. It can be seen that individuals make decisions based on the target value of the option to them.

This study mainly explored the effects of risks and benefits on treatment options in Chinese subjects. This study has the following shortcomings. First, risks and benefits of options were only classified as 36% (low), 64% (medium), and 98% (high). The study found that participants were more likely to accept an option with fewer risks and a higher benefit than an option with fewer risks and a lower benefit. In other words, there is no substantial difference in the probability of moderate or higher benefits being accepted for options with low risks. In order to make behavioral outcomes better differentiated, future studies can divide risks and benefits into multiple levels, such as 10%, 20%, 30%, 90%. The changes in individual treatment compliance under different combinations were further investigated. Second, this study is the first to explore the neural mechanism of the effects of risk and benefits on treatment compliance in Chinese subjects from the perspective of ERP. Future studies are needed to verify whether the detected neural markers are accurate and whether the analysis of psychological significance is reasonable. In addition, Studies have found that prolonged latency of the P300 is associated with slow cognitive processing. Future studies should analyze the latent period data [41].

## 5. Conclusions

During cognitive decision, subjects give priority to the probability of risk. The effect of risk probability was regulated by the cure rate. Moderate risk will receive more attention than low risk, with a larger N1 amplitude. Compared with high risk, low-risk decision integration is easier, and the P300 and LPP amplitude is smaller. While the medium risk obtains more attention early, the LPP amplitude induced by the medium risk is smaller than that of the low risk. The effect of benefit size appears at the late stage of cognitive decision making and adjusts the effect of risk. These results support the satisfaction principle of decision making.

## Figures and Tables

**Figure 1 brainsci-13-00648-f001:**
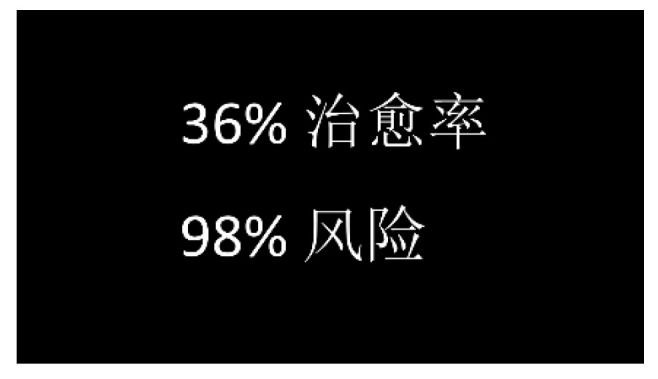
Example of experimental materials.

**Figure 2 brainsci-13-00648-f002:**
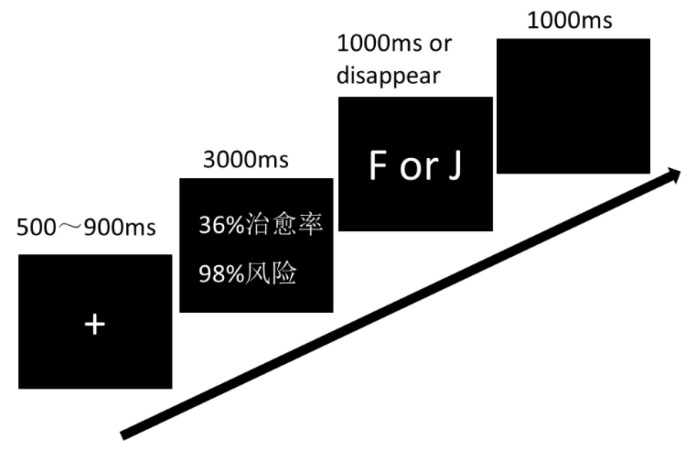
Flow chart of experiment. 治愈率 in English: cure rate; 风险 in English: Risk.

**Figure 3 brainsci-13-00648-f003:**
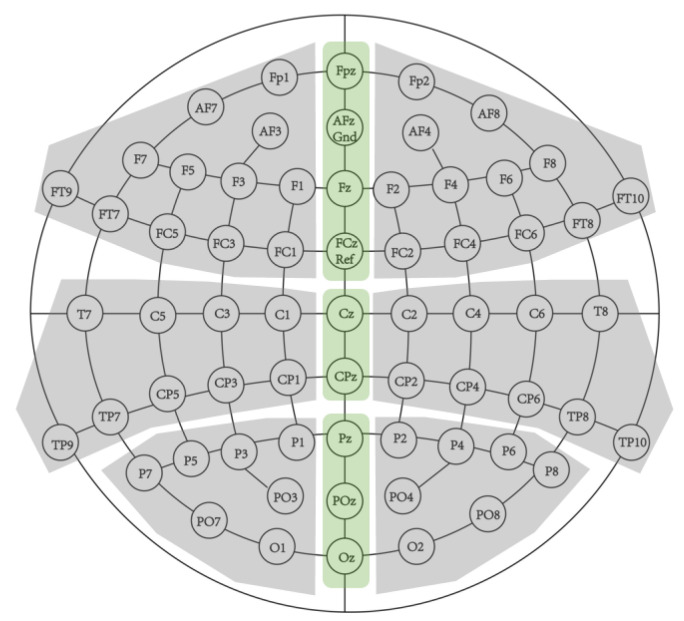
Schema of the topographic electrode clusters. The green area shows the midline clusters. The grey areas represent the lateral clusters.

**Figure 4 brainsci-13-00648-f004:**
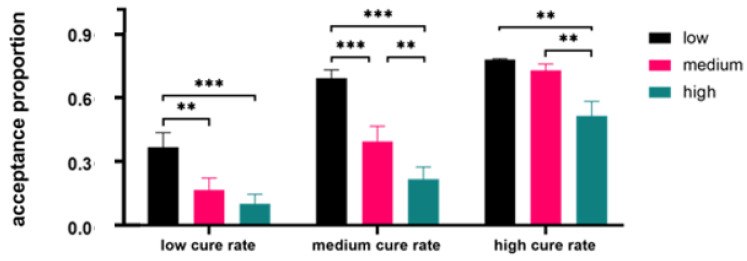
Mean of ‘yes’ responses for the nine conditions and their standard errors (SE). **: *p* < 0.01; ***: *p* < 0.001.

**Figure 5 brainsci-13-00648-f005:**
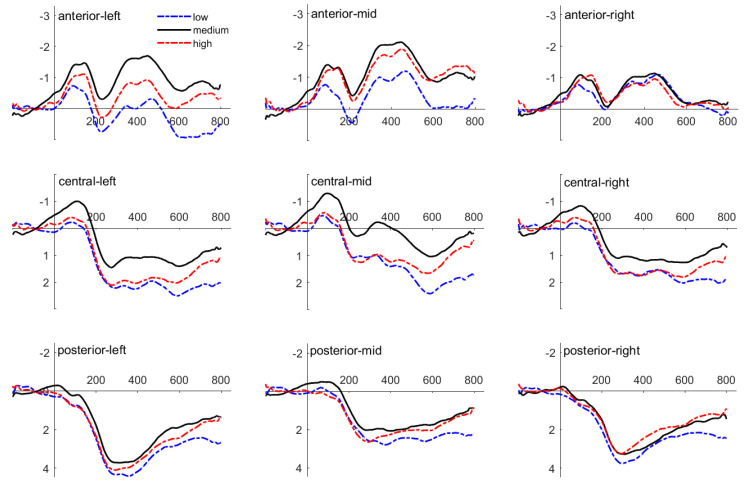
Grand average ERPs plotted for the low-cure rate condition.

**Figure 6 brainsci-13-00648-f006:**
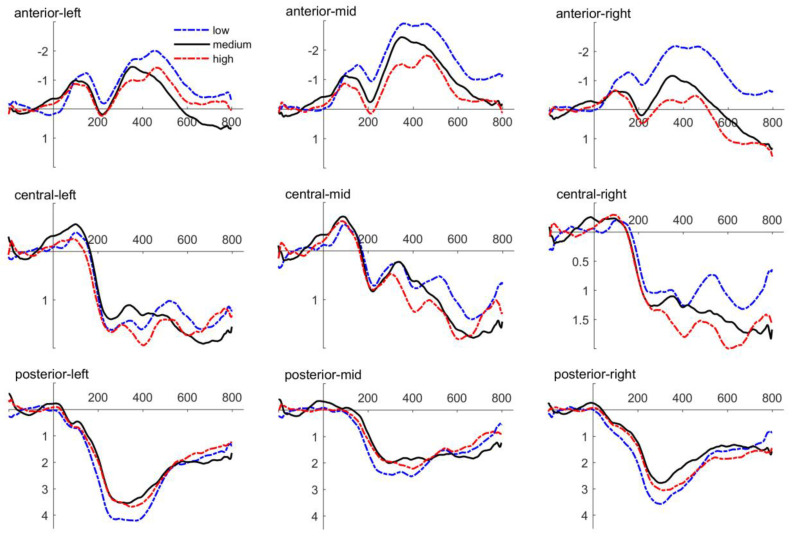
Grand average ERPs plotted for the medium-cure rate condition.

**Figure 7 brainsci-13-00648-f007:**
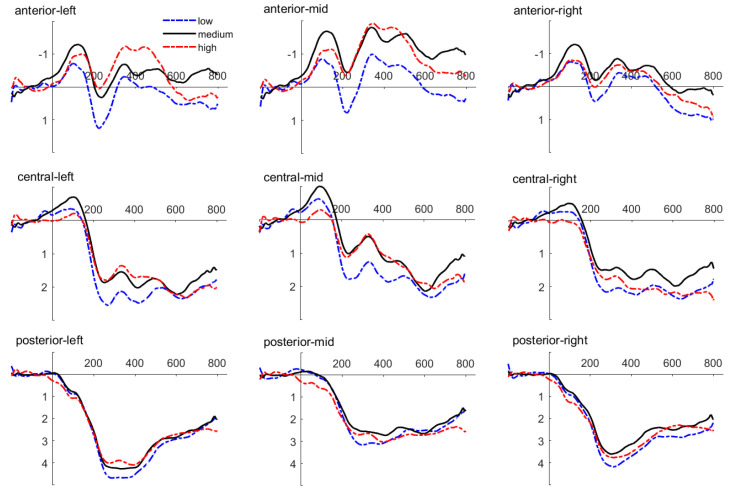
Grand average ERPs plotted for the high-cure rate condition.

**Table 1 brainsci-13-00648-t001:** Descriptive statistics of N1 electrophysiological results.

Anteriority	Laterality	Condition								
		Lc-Lr	Lc-Mr	Lc-Hr	Mc-Lr	Mc-Mr	Mc-Hr	Hc-Lr	Hc-Mr	Hc-Hr
Anterior	Left	0.03	−0.23	−0.08	0.11	−0.16	−0.09	−0.09	−0.15	0.13
	Middle	−0.01	−0.21	−0.03	0.07	−0.23	−0.02	−0.13	−0.16	0.07
	Right	−0.09	−0.15	−0.07	0.06	−0.21	−0.01	−0.17	−0.10	0.02
Central	Left	0.11	−0.20	−0.01	−0.03	−0.10	−0.04	−0.15	−0.11	0.06
	Middle	0.15	−0.26	−0.06	−0.04	−0.18	−0.06	−0.24	−0.14	−0.02
	Right	0.04	−0.14	−0.08	−0.03	−0.13	0.02	−0.15	−0.04	0.004
Posterior	Left	0.21	−0.15	0.08	−0.09	−0.08	0.05	−0.07	−0.01	0.05
	Middle	0.19	−0.19	0.02	−0.07	−0.15	0.03	−0.17	−0.01	0.01
	Right	0.15	−0.06	0.01	−0.03	−0.10	0.05	−0.06	0.03	0.03

Note: Mean values of the event-related potentials (μV). Lc-Lr (low cure rate and low risk), Lc-Mr (low cure rate and medium risk), Lc-Hr (low cure rate and high risk), Mc-Lr (medium cure rate and low risk), Mc-Mr (medium cure rate and medium risk), Mc-Hr (medium cure rate and high risk), Hc-Lr (high cure rate and low risk), Hc-Mr (high cure rate and medium risk), and Hc-Hr (high cure rate and high risk).

**Table 2 brainsci-13-00648-t002:** Descriptive statistics of the P300 electrophysiological results.

Anteriority	Laterality	Condition								
		Lc-Lr	Lc-Mr	Lc-Hr	Mc-Lr	Mc-Mr	Mc-Hr	Hc-Lr	Hc-Mr	Hc-Hr
Anterior	Left	−0.57	−1.12	−0.84	−0.62	−0.83	−0.65	−0.51	−0.98	−0.63
	Middle	−0.67	−1.22	−1.04	−0.89	−0.98	−0.72	−0.71	−1.32	−0.85
	Right	−0.66	−0.86	−0.77	−0.68	−0.49	−0.47	−0.58	−0.89	−0.57
Central	Left	−0.12	−0.73	−0.31	−0.21	−0.45	−0.18	−0.25	−0.45	−0.07
	Middle	−0.39	−1.15	−0.52	−0.43	−0.66	−0.54	−0.60	−0.87	−0.23
	Right	−0.10	−0.51	−0.28	−0.10	−0.16	−0.19	−0.15	−0.29	0.04
Posterior	Left	0.60	0.13	0.61	0.60	0.35	0.48	0.66	0.61	0.87
	Middle	−0.08	−0.46	0.12	0.05	−0.16	0.01	0.02	−0.004	0.51
	Right	0.61	0.36	0.46	0.67	0.32	0.43	0.69	0.59	0.99

Note: Mean values of the event-related potentials (μV). Lc-Lr (low cure rate and low risk), Lc-Mr (low cure rate and medium risk), Lc-Hr (low cure rate and high risk), Mc-Lr (medium cure rate and low risk), Mc-Mr (medium cure rate and medium risk), Mc-Hr (medium cure rate and high risk), Hc-Lr (high cure rate and low risk), Hc-Mr (high cure rate and medium risk), and Hc-Hr (high cure rate and high risk).

**Table 3 brainsci-13-00648-t003:** Descriptive statistics of the LPP electrophysiological results.

Anteriority	Laterality	Condition								
		Lc-Lr	Lc-Mr	Lc-Hr	Mc-Lr	Mc-Mr	Mc-Hr	Hc-Lr	Hc-Mr	Hc-Hr
Anterior	Left	0.38	−0.84	−0.24	−0.76	−0.55	−0.35	0.51	−0.19	−0.48
	Middle	−0.27	−1.20	−0.95	−1.87	−1.32	−0.68	−0.11	−1.13	−1.13
	Right	−0.46	−0.48	−0.51	−1.39	−0.42	0.07	−0.05	−0.53	−0.29
Central	Left	1.62	0.78	1.44	1.21	0.93	1.26	1.80	1.30	1.22
	Middle	1.04	−0.05	1.03	0.44	0.51	0.70	1.45	0.69	0.77
	Right	0.98	0.62	0.95	0.56	0.74	0.83	1.30	1.05	1.17
Posterior	Left	3.58	2.91	3.34	3.58	2.82	2.95	3.88	3.59	3.44
	Middle	2.13	1.55	2.09	2.09	1.49	1.67	2.58	2.19	2.40
	Right	2.95	2.48	2.42	2.86	2.05	2.34	3.31	2.85	3.07

Note: Mean values of the event-related potentials (μV). Lc-Lr (low cure rate and low risk), Lc-Mr (low cure rate and medium risk), Lc-Hr (low cure rate and high risk), Mc-Lr (medium cure rate and low risk), Mc-Mr (medium cure rate and medium risk), Mc-Hr (medium cure rate and high risk), Hc-Lr (high cure rate and low risk), Hc-Mr (high cure rate and medium risk), and Hc-Hr (high cure rate and high risk).

## Data Availability

Data materials can be obtained by contacting the corresponding author.

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
