# Peer review of "The Neural Response Process of Cognitive Decision Making: An ERP Study"

_brainsci, 2023, doi:10.3390/brainsci13040648_

Round 1

Reviewer 1 Report

Comments and Suggestions for Authors

This study investigated how people weigh explicitly stated likelihoods of risk and reward. It shows main effects of both risk and reward on decision making: participants would accept a high risk if the reward is also very likely. There is also an interaction such that increasing the cure-rate moderates the importance of the rate of adverse outcomes.

The ERPs generally track these behavioral effects. There were main effects of risk-rate in the N1 and P300 (no main effect of cure-rate or an interaction). The LPP related to the task in complex ways, with many interactions. When the cure-rate is low, high- and low- risk decisions are easy relative to the medium risk condition, as evidenced by the medium-risk trials eliciting larger ERPs in most ROIs. As the cure-rate increases, the relative difficulty of the risk-assessments also changes. In the medium cure-rate condition, ERPs were more similar across risk-levels than in the low cure-rate condition; however, the low-risk trials tended to elicit larger ERPs.

The study engages with a well-studied question: how to people weigh risk and reward potential when making decisions? They identify a common result: people are risk averse, and will tend to make a decision based on one dimension (risk or reward) that is exceedingly likely (or unlikely), and will integrate both dimensions only as necessary. Integrating two dimensions of information requires more effort, which is reflected in the ERPs. This same effect can be seen in basic categorization studies, and is a ubiquitious psychological phenomenon.

The linking hypotheses between evaluating evidence in this task and the N1, P300, and LPP strike me as weak, or at least underdeveloped. The ERP component of the study felt very exploratory. This analysis was overwhelming and without clear hypotheses about the neural data it didn't leave an impression on me.

The study itself involves presenting participants with individual trials where a risk-reward probability pair was presented. Participants decided whether to except each treatment, in isolation from other options. This paradigm seems to have important weaknesses. Participants are told to imagine they have a "serious disease", but no details are given about the prognosis. Without establishing a baseline for what people expect their outcome to be if they abstain from treatment, how do the participant's judgments make sense? I may be willing to accept a high risk if my prognosis is that I have days or hours to live without treatment.

Building on this... the risk is "deterioration or even death". Is a 30% chance of death really considered low risk?

Also, participants are given a pair of probabilities to compare, but the authors themselves note that naturalistic decision making is not a rational comparision of probabilties.

These, among other issues, make it difficult for me to read too much into behavior on this task.

Also, the reporting of data for the neuroimaging, in three enormous tables, is fine for archival but not effective for conveying findings.

When a figure is displaying probabilities, the y-axis should not exceed the valid range for a probability.

Reviewer 2 Report

Comments and Suggestions for Authors

This is an interesting paper. Some recommendations for this revision:

Please include the clinical role of P300 and compare and contrast your findings related to its use in older adults as well (see indicative references below):

Olichney, J., Xia, J., Church, K. J., & Moebius, H. J. (2022). Predictive power of cognitive biomarkers in neurodegenerative disease drug development: Utility of the P300 event-related potential. Neural Plasticity2022.

Gao, L., Gu, L., Shu, H., Chen, J., Zhu, J., Wang, B., ... & Zhang, Z. (2021). The reduced left hippocampal volume related to the delayed P300 latency in amnestic mild cognitive impairment. Psychological Medicine51(12), 2054-2062.

In addition to that, brain volumes have been found to be involved in decision-making regarding finances (Giannouli, V., & Tsolaki, M. (2019). Are left angular gyrus and amygdala volumes important for financial capacity in mild cognitive impairment?. Hellenic journal of nuclear medicine22, 160-164.). You could discuss this point regarding your findings about healthcare decisions in young adults with EEG:

Fang C, Zhang Y, Zhang M, Fang Q. P300 Measures and Drive-Related Risks: A Systematic Review and Meta-Analysis. Int J Environ Res Public Health. 2020 Jul 22;17(15):5266.

The introduction needs to include more references regarding in general decision-making across the life span and the Methods section has to clarify how the participants were screened (were they healthy, what neuropsychological tests were administered in order to check for that?).

The materials sections needs to be described in more detail (the reader does not understand what stimuli were used exactly).

The discussion is brief. Please extend on the main findings.

Round 2

Reviewer 1 Report

Comments and Suggestions for Authors

Thank you for your thoughtful responses to my comments in the first round. I appreciate the revisions you have made. I still find the mapping of cognitive processes to ERPs as described in the paper to be overstating the case. These components are not so easily described. Also, my thoughts on the experiment design have not changed. I feel that the way that decisions are being made, the ways that information is limited in some ways and unnaturally explicit in others, impede direct generalization to risk/reward analyses in real medical situations.

I do not have further comments.

Reviewer 2 Report

Comments and Suggestions for Authors

The questions made by both reviewers have not been addressed adequately by the author(s) as the indicated changes in the manuscript (red highlighted text shows). No fundamental change has been made regarding the review of the literature and the discussion.
